# Screening for Metallo-Beta-Lactamases Using Non-Carbapenem Agents: Effective Detection of MBL-Producing *Enterobacterales* and Differentiation of Carbapenem-Resistant *Enterobacterales*

**DOI:** 10.3390/antibiotics12071146

**Published:** 2023-07-03

**Authors:** Kentarou Takei, Hajime Kanamori, Asami Nakayama, Mikiko Chiba, Yumiko Takei, Issei Seike, Chiho Kitamura, Hiroaki Baba, Kengo Oshima, Koichi Tokuda

**Affiliations:** 1Department of Infectious Diseases, Internal Medicine, Tohoku University Graduate School of Medicine, Sendai 980-8574, Japan; kanamori@med.tohoku.ac.jp (H.K.); chiho.kitamura.d1@tohoku.ac.jp (C.K.); koshima@med.tohoku.ac.jp (K.O.);; 2Department of Laboratory Medicine, Tohoku University Hospital, Sendai 980-8574, Japan

**Keywords:** ceftazidime, cefoperazone/sulbactam, carbapenem-resistant *Enterobacterales*, metallo-β-lactamase

## Abstract

Metallo-beta-lactamases (MBLs) are enzymes that break down carbapenem antibiotics, leading to carbapenem-resistant organisms. Carbapenemase-resistant *Enterobacterales* (CRE) is one of them. Outbreaks of CRE infection can occur in healthcare facilities and lead to increased deaths, illness, and medical costs. This study was conducted to detect MBLs using non-carbapenem agents and exclude MBLs among CRE isolates. A total of 3776 non-duplicate sequential *Enterobacterales* isolates from a single facility were screened between January 2019 and December 2022 using non-carbapenem agents, ceftazidime and cefoperazone/sulbactam. Positive 153 isolates (4.0%) were further tested using carbapenemase-confirmation tests and verified through polymerase chain reaction (PCR) testing. Fifteen imipenemase (IMP)-type MBL-producing *Enterobacterales* (0.4%) including one susceptible to carbapenems were identified. Moreover, 160 isolates (4.2%) meeting the criteria for CRE were directly subjected to PCR testing. All fourteen CRE isolates with MBLs identified through PCR testing were found to be the same strains screened using ceftazidime and cefoperazone/sulbactam. Screening using ceftazidime and cefoperazone/sulbactam can effectively detect MBL-producing *Enterobacterales* strains. This screening method showed comparable results to screening with meropenem, potentially serving as a supplementary approach and contributing to differentiating between MBL- and non-MBL-producing CRE strains. Our findings support these screening methods, particularly in regions where IMP-type MBLs are prevalent.

## 1. Introduction

Metallo-beta-lactamase (MBL) is a type of carbapenemase enzyme that breaks down carbapenem antibiotics. MBLs are zinc-type β-lactamases categorized under Class B according to the Ambler classification, the most widely used classification of β-lactamases [1]. *Enterobacterales* that produce carbapenemase are referred to as carbapenemase-producing *Enterobacterales* (CPE). Clinical MBL-producing *Enterobacterales* was first reported in 1994 in a *Serratia marcescens* strain resistant to imipenem [2]. Resistance of MBL-producing *Enterobacterales* to carbapenem drugs is of particular concern as they have the potential to horizontally transmit resistance to other *Enterobacterales* via conjugative transfer mediated by plasmids [3]. Following the emergence of new types of carbapenemase-producing bacteria in the United States, such as the *Klebsiella pneumoniae* carbapenemase-producing strains outbreak in 1996 and the New Delhi metallo-beta-lactamase-producing strains in 2009, the Centers for Disease Control and Prevention issued a warning about carbapenem-resistant *Enterobacterales* (CRE) in 2013, drawing attention to the presence of CRE [4,5,6]. Outbreaks of CRE infection frequently occur in healthcare facilities and lead to increased deaths, illness, and medical costs [7]. In Japan, the definition of CRE within the country was established in September 2014 under the Infectious Diseases Control Law, which includes the presence of *Enterobacterales* isolates with a minimum inhibitory concentration (MIC) ≥2 µg/mL for meropenem, ≥2 µg/mL for imipenem, and ≥64 µg/mL for cefmetazole. As CRE is defined by resistance to carbapenem drugs, it encompasses various mechanisms of resistance, including resistance due to MBLs. CPEs account for 16.5–28% of CRE isolates; in Japan, most of these CPEs were reported to have imipenemases (IMP)-type MBLs [8]. However, MBL-producing *Enterobacterales* do not necessarily correspond to CRE, as those strains may be susceptible to carbapenem.

Representative clinical laboratory organizations, such as the Clinical and Laboratory Standards Institute (CLSI) or European Committee on Antimicrobial Susceptibility Testing (EUCAST), have proposed as methods for CPE screening [9,10]. A common aspect of these testing methods is the evaluation of drug susceptibility, particularly toward carbapenem agents such as meropenem. Nishio et al. [11] conducted a survey of MBL-producing Gram-negative rods from 13 clinical laboratories in a specific region of Japan, in 2004, ten years before the Infectious Diseases Control Law regarding CRE was established in Japan. As a first-step screening for the collected bacterial strains, drug susceptibility testing using ceftazidime and cefoperazone/sulbactam was employed. Then, confirmation of carbapenemase production in drug-resistant strains was carried out using carbapenem confirmation tests and PCR examinations, as reported in the study. In our facility, carbapenemase confirmation tests were previously conducted on suspicious MBL-producing *Enterobacterales*, following preset protocols provided by the manufacturer. These protocols involved assessing non-susceptibility to third- or fourth-generation cephalosporins and carbapenem agents. Additionally, empirical judgment was employed by observing resistance to imipenem. However, this method often imposed a large burden in terms of the laboratory workload. After that, focusing on screening with ceftazidime and cefoperazone/sulbactam, we have been implementing MBL screening methods with these two drugs for some time as part of the testing protocol.

In this study, we evaluated the usefulness of screening for MBLs using non-carbapenem antimicrobial agents, ceftazidime and cefoperazone/sulbactam, against *Enterobacterales*. Drug susceptibility testing for ceftazidime and cefoperazone/sulbactam among CRE, facilitated by automated antimicrobial susceptibility testing, demonstrates the ability to rapidly identify non-MBL-producing CRE.

## 2. Results

### 2.1. Total Bacterial Isolates and Those Screened Using Ceftazidime and Cefoperazone/Sulbactam

A total of 3776 non-duplicate sequential strains were obtained (Appendix A). Of these, a total of 153 isolates (4.0%) were screened using ceftazidime and cefoperazone/sulbactam. A list of those strains is shown in Table 1.

### 2.2. Enterobacterales Screened Using Ceftazidime and Cefoperazone/Sulbactam

Of the 153 isolates screened using ceftazidime and cefoperazone/sulbactam, *Enterobacter cloacae* was predominant, accounting for the highest proportion (40.5%) within this population, followed by *K. pneumoniae* (15.7%), *Escherichia coli* (15.0%), and *Citrobacter braakii* (11.8%). *Klebsiella aerogenes* was observed in only four cases. Of the 153 isolates, 15 (9.8%) were selected after the sodium mercaptoacetic acid (SMA) test (Figure 1). These fifteen strains tested positive in PCR examination, and all were identified as IMP-1 type MBL. Of these, fourteen MBL-possessing strains resistant to meropenem were observed, which corresponded to CRE. However, one isolate was identified as carbapenem-sensitive MBL, showing susceptibility to both meropenem and imipenem. The MBLs-producing isolates are listed in Table 2.

### 2.3. Carbapenem-Resistant Enterobacterales

A total of 160 isolates were identified as CREs, corresponding to approximately 4.2% of the initial 3776 strains (Table 3). Among these isolates, 5 (3.1%) were resistant only to meropenem, 140 (87.5%) were resistant only to imipenem, and 15 (9.4%) were resistant to meropenem and imipenem. The major strains of CRE were *K. aerogenes* and *E. cloacae* (Table 3). The proportions of identified CRE strains among *K. aerogenes* and *E. cloacae* were 44.2% (96/217) and 12.4% (54/435), respectively. All CRE of *K. aerogenes* lacked MBLs and were resistant only to imipenem. Excluding isolates possessing MBLs, 95% (42/44) of *E. cloacae* showed resistance only to imipenem. Among the CRE strains, 14 strains (8.8%) produced MBLs, as confirmed via PCR testing. Among the CRE isolates that exhibited resistance solely to imipenem, none had MBLs; however, all MBL-producing CRE strains were resistant to meropenem.

### 2.4. Carbapenem, Ceftazidime, and Cefoperazone/Sulbactam-Resistant Enterobacterales

All MBL-producing CREs resistant to both ceftazidime and cefoperazone/sulbactam (Table 2). Therefore, in cases where susceptibility was demonstrated to either of the two drugs, potential MBL-producing CREs could be ruled out with 100% sensitivity and with statistical significance (*p* < 0.001) using Fisher’s exact test (Appendix A).

## 3. Discussion

We focused on two main aspects: screening of MBLs-producing *Enterobacterales* using non-carbapenem agents and exploring the potential of using drug susceptibility to ceftazidime and cefoperazone/sulbactam in order to rapidly exclude MBL-producing *Enterobacterales* among CRE isolates.

We screened on 3776 clinical specimens using ceftazidime and cefoperazone/sulbactam and identified 1 isolates of carbapenem-susceptible MBL. The EUCAST method is a highly effective approach and is likely to become a standard indicator; however, even with CPE screening based on the EUCAST criteria, 1.6% cases cannot be screened [12]. This study is a rare investigation focusing on the screening of MBLs using non-carbapenem agents. Screening with ceftazidime and cefoperazone/sulbactam proved to be a method as equally effective as screening with meropenem, at least with an MIC for meropenem >0.25 µg/mL; however, it is unknown whether our isolate of carbapenem-susceptible MBL indicating the MIC for meropenem ≤0.25 μg/mL can be screened using the EUCAST criteria due to lack of MIC details for meropenem.

CLSI and EUCAST, as world-renowned testing organizations, have proposed breakpoints for screening CPE. Each organization recommends conducting carbapenemase confirmation tests when the MIC of imipenem is ≥2 μg/mL and that of meropenem is ≥0.25 μg/mL when screening for CPE [9,10,13]. However, susceptibility to meropenem is given greater emphasis over that to imipenem [9,10]. Indeed, in our study, no MBL-producing CREs were identified through screening using only imipenem. Hence, screening for CPE using imipenem may have low utility potential. It was reported that at the meropenem screening breakpoint provided by CLSI (≥2 μg/mL), 86.2% of CPE isolates would have been detected. In contrast, the EUCAST screening breakpoint for meropenem (≥0.25 μg/mL) resulted in detection of 98.4% of CPE isolates [12]. The meropenem screening breakpoint based on EUCAST criteria, which is also included in the joint proposal from four domestic academic societies (Japanese Society of Chemotherapy, Japanese Association for Infectious Diseases, Japanese Society for Infection Prevention and Control, and Japanese Society for Clinical Microbiology), was suggested for screening MBLs [14]. However, while these criteria are academically recommended, it is unclear whether panels satisfying these recommendations are widely being used in actual clinical laboratories.

This study also investigated MBLs from the perspective of CRE. Among the CRE strains, *K. aerogenes* accounted for 59.6%, followed by *E. cloacae*, which is consistent with previous reports in Japan [8]. In our study, all *K. aerogenes* strains were resistant solely to imipenem and non-MBL-producers. Even in *E. cloacae*, 95% of non-MBL-producing CREs were resistant only to imipenem. Among the non-MBL-producing CRE strains, CRE strains of *K. aerogenes* and *E. cloacae* have developed via mechanisms such as AmpC overexpression and efflux pumps, and downregulation of porin is involved in the resistance mechanisms, including the coexistence of extended-spectrum β-lactamase (ESBLs) [15,16,17,18]. Screening based on the definition of CRE more frequently detects AmpC-producing bacteria (e.g., *K. aerogenes* and *E. cloacae*). However, screening with ceftazidime and cefoperazone/sulbactam tends to be more effective in detecting ESBL-producing bacteria (e.g., *E. coli* and *K. pneumoniae*) [19,20]. Finally, screening with ceftazidime and cefoperazone/sulbactam excluded 98.2% (213/217) of *K. aerogenes* and 85.7% (373/435) of *E. cloacae*, which are representative AmpC-producing bacteria. Additionally, it excluded 98.2% (1240/1263) of *E. coli* and 96.3% (629/653) of *K. pneumoniae*, which are representative ESBLs, through screening.

In our current study, 14 isolates of MBL-producing CREs showed resistance to ceftazidime and cefoperazone/sulbactam. In contrast, no MBLs were detected using PCR testing in the CRE strains that exhibited susceptibility to either ceftazidime or cefoperazone/sulbactam. Based on these findings, all CRE strains demonstrating susceptibility to ceftazidime or cefoperazone/sulbactam may be non-MBL strains, as confirmed through statistical analyses. This finding suggests the possibility of MBLs being excluded, potentially contributing to early infection control based on automated drug susceptibility testing. It is important to differentiate CPE from non-CPE strains among CRE because the CPE acquisition is associated with horizontal transmission, whereas non-carbapenemase-producing CRE is associated with carbapenem exposure. Differences in the acquisition factors necessitate tailored infection prevention efforts [21].

This study has several limitations. The molecular epidemiology of carbapenemase varies widely worldwide, and the prevalence of IMP-type MBL is centered around east Asia or other countries, including Japan [22]. Particularly, the IMP type is predominant in Japan [8]. The molecular epidemiology differs from that in other countries [23]. We focused on bacterial strains from a single facility in a specific region; hence, our results may not reflect the global situation. In addition, it is unclear whether screening with ceftazidime and cefoperazone/sulbactam would be effective against other CPE strains because all of our MBLs were of the IMP type. Second, carbapenem confirmation tests were not performed on all the strains that were excluded during the screening process of ceftazidime and cefoperazone/sulbactam. Therefore, we could not rule out the possibility of the presence of MBLs or other CPE strains that are susceptible to carbapenem among the negative strains identified through screening with these two agents.

## 4. Materials and Methods

### 4.1. Study Design for Bacterial Isolates

A total of 3776 non-duplicate sequential *Enterobacterales* isolates were cultured and identified from various clinical specimens at Tohoku University Hospital (1160 beds) located in Sendai, which is in the northeastern part of Japan, between January 2019 and December 2022. These *Enterobacterales* strains were identified using a VITEK^®^ MS (Sysmex-bioMérieux, Tokyo, Japan). Only the first 3839 cultured non-duplicate sequential isolates were retrospectively selected. These isolates were initially screened based on their drug susceptibility to ceftazidime and cefoperazone/sulbactam. Isolates that screened positive underwent further screening using the SMA method, which is a well-known carbapenemase confirmation test [24]. Finally, PCR testing was performed on strains that tested positive in the SMA method. However, within the first cultured non-duplicate sequential isolates, bacteria that met the criteria for CRE were selected and directly tested using PCR. The results of PCR testing were compared with those of the screening method using ceftazidime and cefoperazone/sulbactam.

### 4.2. Antimicrobial Susceptibility Testing

Drug susceptibility testing of the bacterial isolates was performed using Microscan WalkAway^®^ (Beckman Coulter, Brea, CA, USA) and accompanying panels of the Microscan Neg^®^ series (Beckman Coulter, Brea, CA, USA). The bacterial solution used for this test was mainly prepared using the prompt inoculation method, but the standard inoculation method (CLSI method) was used for preparing potential CRE isolates or blood specimens [25,26,27]. Because we used a previous commercial panel from an automated system, the MICs for each drug were determined according to the provided manufacturer’s instructions and were as follows: meropenem, ≤0.25 to >2 μg/mL; imipenem, ≤0.5 to >2 μg/mL; ceftazidime, ≤1 to >8 µg/mL; cefoperazone/sulbactam, ≤8 to >32 μg/mL; cefmetazole, ≤4 to >32 µg/mL; ceftriaxone, ≤0.5 to >2 μg/mL; cefepime, ≤1 to >16 µg/mL; levofloxacin, ≤0.12 to >4 µg/mL; tazobactam/piperacillin, ≤4 to >64 µg/mL, and gentamicin, ≤2 to >8 µg/mL.

### 4.3. Definition and Breakpoint of Screening Antibiotics

The criteria for drug susceptibility were based on the CLSI guidelines, with MIC values of ceftazidime ≥8 μg/mL. However, the CLSI has not established breakpoints for cefoperazone/sulbactam in *Enterobacterales*. Therefore, referring to the breakpoint of cefoperazone in past studies, the cut-off MIC for cefoperazone/sulbactam was set to ≥32 μg/mL [11,28]. When a bacterial strain showed resistance to both agents, it was defined as a positive result. CRE isolates were identified based on criteria defined by the Japanese Ministry of Health, Labour and Welfare, which defines CRE as having an MIC ≥ 2 μg/mL for meropenem, MIC ≥ 2 μg/mL for imipenem, and MIC > 32 μg/mL for cefmetazole.

### 4.4. Identification of Metallo-Beta-Lactamase

The carbapenemase confirmation test was performed using the SMA test with an SMA Disk EIKEN^®^ (Eiken Chemical Co. Ltd., Tokyo, Japan), according to the manufacturer’s instructions. The tested bacteria were streaked onto Mueller–Hinton agar using a cotton swab after being inoculated and adjusted to the same turbidity as the McFarland standard of 0.5 in sterile saline solution. Two CAZ disks were placed on the agar, at least 3 cm apart from each other. An SMA disk was placed approximately 1.5–2 cm away from the center of one of the CAZ disks. Within 15 min, the plates were placed in a 35 °C incubator and cultured for 16–18 h. The inhibition zones were measured after incubation. If the inhibition zone of CAZ adjacent to the SMA disk expanded by ≥5 mm perpendicular to the axis connecting the centers of the SMA and CAZ disks, the tested bacteria were considered as likely producers of MBL. If no inhibition zone was observed using the SMA disk method, an imipenem disk was used instead. For all positive isolates in the SMA test and for all CRE isolates, subsequent PCR testing was performed. Carbapenemase genes in the CRE isolates were identified using a Cica Geneus^®^ Carbapenemase Genotype Detection KIT2 (Kanto Chemical Co., Tokyo, Japan), which can detect the following genes: IMP-1, VIM, GES, KPC, NDM, OXA-48, and IMP-6. Bacterial processing and thermal cycling were performed according to the manufacturer’s instructions.

### 4.5. Statistical Analysis

Fisher’s exact test was used to compare the proportion between two independent groups. Sensitivity and specificity analyses were performed using common methods. The statistical significance of sensitivity and specificity was analyzed using R statistical software (version 4.1.2; R Foundation for Statistical Computing, Vienna, Austria) with a significance level of *p* < 0.05.

## 5. Conclusions

In conclusion, we demonstrated that screening with ceftazidime and cefoperazone/sulbactam, as non-carbapenem agents, can effectively detect MBL-producing *Enterobacterales* strains, including those susceptible to meropenem. The EUCAST method is a highly effective approach; however, this screening method showed comparable results to screening with meropenem and may serve as a supplementary approach, even for MBL strains with a meropenem MIC ≤ 0.25 μg/mL. Additionally, the use of ceftazidime and cefoperazone/sulbactam for screening showed utility in excluding non-MBL-producing CRE strains. Therefore, the ceftazidime and cefoperazone/sulbactam screening method can be effectively utilized, particularly in regions where IMP-type MBLs are predominant.

## Figures and Tables

**Figure 1 antibiotics-12-01146-f001:**
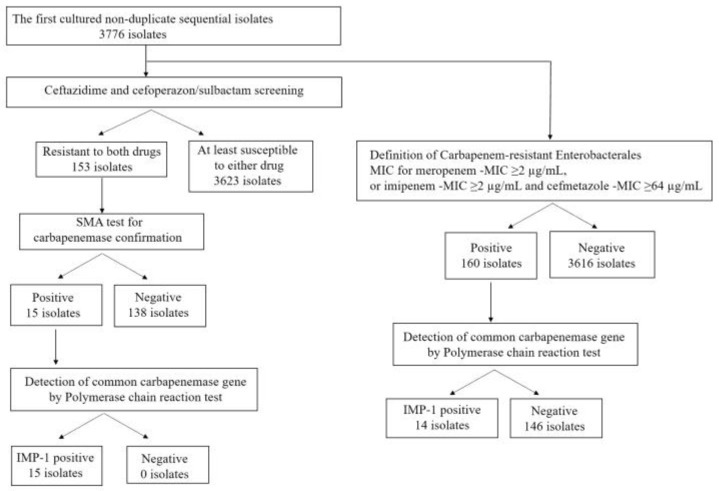
Identification and exclusion process for isolates screened using ceftazidime and cefoperazone/sulbactam, based on the definition of carbapenem-resistant *Enterobacterales*. Abbreviations: IMP, imipenemase; SMA, sodium mercaptoacetic acid; MIC, minimum inhibitory concentration.

**Table 1 antibiotics-12-01146-t001:** List of *Enterobacterales* resistant to both ceftazidime and cefoperazone/sulbactam among the first cultured non-duplicate sequential isolates.

*Enterobacterales* Resistant to Ceftazidime and Cefoperazone/Sulbactam	n = 153
*Enterobacter cloacae*	62
*Klebsiella pneumoniae*	24
*Escherichia coli*	23
*Citrobacter braakii*	18
*Serratia marcescens*	6
*Klebsiella aerogenes*	4
*Citrobacter* sp.	3
*Klebsiella oxytoca*	2
*Morganella morganii*	2
*Proteus mirabilis*	2
*Serratia liquefaciens*	2
*Citrobacter werkmanii*	1
*Citrobacter youngae*	1
*Enterobacter* sp.	1
*Hafnia alvei*	1
*Yersinia enterocolitica*	1

**Table 2 antibiotics-12-01146-t002:** Characteristics and drug susceptibility of MBL-producing isolates.

				Antibiotic Minimum Inhibitory Concentration (μg/mL)									
No.	Bacterial Strains	CRE or Non-CRE	Sample	MEM	IPM	CMZ	CAZ	CFP/SBT	CRO	FEP	LVX	TZP	GM
1	*Enterobacter cloacae*	CRE	Urine	>2	2	>32	>8	>32	>2	>16	1	≤4	≤2
2	*Enterobacter cloacae*	CRE	Sputum	>2	>2	>32	>8	>32	>2	>16	4	64	≤2
3	*Enterobacter cloacae*	CRE	Blood	2	≤0.5	>32	>8	32	>2	>16	1	8	≤2
4	*Enterobacter cloacae*	CRE	Blood	>2	>2	>32	>8	>32	>2	16	1	4	≤2
5	*Enterobacter cloacae*	CRE	Urine	>2	>2	>32	>8	>32	>2	16	≤0.12	16	≤2
6	*Enterobacter cloacae*	CRE	Urine	>2	>2	>32	>8	>32	>2	16	1	>64	≤2
7	*Enterobacter cloacae*	CRE	Urine	2	1	>32	>8	>32	>2	8	≤0.12	≤4	≤2
8	*Enterobacter cloacae*	CRE	Sputum	>2	1	>32	>8	32	>2	8	1	16	≤2
9	*Enterobacter cloacae*	CRE	Urine	>2	2	>32	>8	>32	>2	>16	2	>64	≤2
11	*Enterobacter cloacae*	CRE	Sputum	>2	2	>32	>8	>32	>2	>16	1	≤4	≤2
10	*Enterobacter cloacae*	Non-CRE	Sputum	≤0.25	≤0.5	>32	>8	>32	>2	8	1	16	≤2
12	*Klebsiella pneumoniae*	CRE	Bile	>2	>2	>32	>8	>32	>2	>16	4	64	≤2
13	*Klebsiella pneumoniae*	CRE	Puncture fluid	>2	>2	>32	>8	>32	>2	>16	≤0.12	64	≤2
14	*Klebsiella pneumoniae*	CRE	Urine	>2	>2	>32	>8	>32	>2	>16	1	>64	≤2
15	*Escherichia coli*	CRE	Urine	2	2	>32	>8	32	>2	4	1	≤4	≤2

Abbreviations: MEM, meropenem; IPM, imipenem; CMZ, cefmetazole; CAZ, ceftazidime; SBT/CFP, cefoperazone and sulbactam; CRO, ceftriaxone; FEP, cefepime; LVX, levofloxacin; TZP, tazobactam/piperacillin; GM, gentamicin; MBL, metallo-beta-lactamase; CRE, carbapenemase-resistant *Enterobacterales.*

**Table 3 antibiotics-12-01146-t003:** Carbapenem-resistant *Enterobacterales* among 3776 strains between January 2019 and December 2022.

Carbapenem-Resistant *Enterobacterales*	n = 160
*Klebsiella aerogenes*	96
*Enterobacter cloacae*	54
*Klebsiella pneumoniae*	8
*Escherichia coli*	2

## Data Availability

The datasets of the current study are not publicly due to the fact that they contain a great deal of detailed patient’s information. The dataset is owned by the Department of Infectious Diseases, Internal Medicine, Tohoku University Graduate School.

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
