# Peer review of "Screening for Metallo-Beta-Lactamases Using Non-Carbapenem Agents: Effective Detection of MBL-Producing Enterobacterales and Differentiation of Carbapenem-Resistant Enterobacterales"

_antibiotics, 2023, doi:10.3390/antibiotics12071146_

Round 1
Reviewer 1 Report
In general, the manuscript is well-presenting and provides an interesting methodology to identify MBL strains in Enterobacterales. However, some corrections are needed in the current version of this manuscript, mainly on redaction and the presentation of the different sections. Those are commented by sections as follows:
Abstract
The results obtained from the analysis are not well described in the abstract; they also introduce an abbreviation at the end that was not explained (IMP-type MBLs).
Introduction
Line 64, replace the number “10” by word in the text.
Line 74, fix the word “Enterobacterales”.
Results
In general, this section needs major revision by the authors. The redaction in several parts is confusing. It would be better if the authors separated well de MBL from CRE. The initial number of 7058 strains maybe is not necessary to be provided due to the final number of strains tested was 3839.
For lines 81 to 83, revise the redaction when the authors describe 154 strains and then 161 with CRE is confusing. Explain better that those 154 were representative of finding MBL.
Table 1 could be separated in two, one for the 154 strains and the other for the 161 strains. In addition, the Aeromonas genus is not an Enterobacterales bacteria (taxonomy).
Lines 108 and 111, revise the redaction and why the authors use the word cases for those strains that represent MBL is confusing.
Figure 1 has several mistakes. First, fix the box when appears “152”. Then the authors present the PCR results only as positive or negative. This makes questions to the readers like positive to what? Please be clearer. Moreover, the figure showed things like SMA testing for looking for MBL strains and the definition of CRE; this should be clearly explained in the figure legend.
Figure 2 is unnecessary due to in the text and Table 2 is reflected the results.
Discussion
The redaction of this section needs to be revised too. For example, lines 148 to 151 are redundant. Lines 153 and 154 are confusing. In general, all the paragraphs from lines 148 to 156 should be fixed.
Materials and methods
In line 229, the authors refer to an 1100-bedded hospital. What if the authors put the region, city or the exact hospital where the strains were collected?
Line 261, “MIC MIC” is correct? Please revise.
Author Response
Point-by-point response to the Reviewers’ comments
Thank you for taking the time to review and provide guidance on my paper. We have revised the content of the paper based on your valuable comments. We would like to request another review of the manuscript. Thank you for your assistance.
- Abstract
Reviewer's suggestion or questions
The results obtained from the analysis are not well described in the abstract; they also introduce an abbreviation at the end that was not explained (IMP-type MBLs).
Response:
Thank you for your suggestion regarding the abbreviations in the abstract. The necessary revisions have been made. Additionally, we have included detailed results in the abstract. Thank you for pointing this out.
- Introduction
Reviewer's suggestion or questions
2-1. Line 64, replace the number “10” by word in the text.
Response:
The numeral "10" has been changed to "ten" as suggested.
Reviewer's suggestion or questions
2.2 Line 74, fix the word “Enterobacterales”.
Response:
This term has been corrected according to your comment.
- Results
Reviewer's suggestion or questions
3-1 In general, this section needs major revision by the authors. The redaction in several parts is confusing. It would be better if the authors separated well de MBL from CRE. The initial number of 7058 strains maybe is not necessary to be provided due to the final number of strains tested was 3776.
Response:
Thank you for pointing out the confusion caused by the mix of MBL and CRE. As per your suggestion, we have moved the details on CRE strains to section 2.3, specifically under "Carbapenem-resistant Enterobacterales," and summarized the information there. Furthermore, I have removed the mention of 7058 strains throughout the paper. The 3839 strains included bacteria outside the Enterobacteriaceae family. Therefore, we have revised this value to 3776 strains, focusing specifically on Enterobacteriaceae bacteria.
Reviewer's suggestion or questions
3-2 For lines 81 to 83, revise the redaction when the authors describe 154 strains and then 161 with CRE is confusing. Explain better that those 154 were representative of finding MBL.
Response:
As mentioned in our previous response, we have separated the sections for MBL and CRE. We have made the necessary revisions to ensure a clear distinction between these two topics.
Reviewer's suggestion or questions
3-3 Table 1 could be separated in two, one for the 154 strains and the other for the 161 strains. In addition, the Aeromonas genus is not an Enterobacterales bacteria (taxonomy).
Response:
As suggested, we have split the table for the 154 strains screened with two agents and 161 strains of CRE. We apologize for the oversight of including Aeromonas genus. We have double-checked the number of strains for each category, which has been corrected to 153 and 160 strains, respectively. Thank you for pointing this out. We apologize for any confusion.
Reviewer's suggestion or questions
3-4 Lines 108 and 111, revise the redaction and why the authors use the word cases for those strains that represent MBL is confusing.
Response:
We understand that the use of "Case" may be a matter of language nuance. As suggested, we have made the necessary revisions and replaced "Case" with "strain" or "isolate," which is more appropriate in this context. Thank you for helping us to ensure the accuracy of the terminology.
Reviewer's suggestion or questions
3-5 Figure 1 has several mistakes. First, fix the box when appears “152”. Then the authors present the PCR results only as positive or negative. This makes questions to the readers like positive to what? Please be clearer. Moreover, the figure showed things like SMA testing for looking for MBL strains and the definition of CRE; this should be clearly explained in the figure legend.
Response:
We have corrected the number 152. Furthermore, as you pointed out, the reader may be confused regarding the positive or negative results of PCR. Therefore, we have included the phenotype of the detected carbapenemases to improve clarity. Additionally, we have revised the legends for ceftazidime and cefoperazone/sulbactam, as well as the SMA test, to include more specific explanations and improve clarity.
Reviewer's suggestion or questions
3-6 Figure 2 is unnecessary due to in the text and Table 2 is reflected the results.
Response:
We agree with removing the figure and table, as this information is presented in the written content.
- Discussion
Reviewer's suggestion or questions
4-1 The redaction of this section needs to be revised too. For example, lines 148 to 151 are redundant. Lines 153 and 154 are confusing. In general, all the paragraphs from lines 148 to 156 should be fixed.
Response:
We wanted to remind readers of the focus of the research before discussing each section. However, as you mentioned, this may be redundant. Therefore, we have removed this information to avoid unnecessary repetition of the introductory text.
- Materials and methods
Reviewer's suggestion or questions
5-1 In line 229, the authors refer to an 1100-bedded hospital. What if the authors put the region, city or the exact hospital where the strains were collected?
Response:
We have included specific details such as the geographical location, hospital names, and hospital size. We appreciate your effort in ensuring the accuracy of the bed numbers and have made the necessary corrections.
Reviewer's suggestion or questions
5-2 Line 261, “MIC MIC” is correct? Please revise.
Response:
This error has been corrected.
Reviewer 2 Report
Congratulations, it is a very nice study,with direct practical importance.
Only a few uncertain formulations detected
See the attachment.

Author Response
Point-by-point response to the Reviewers’ comments
Thank you for taking the time to review and provide guidance on my paper. We have revised the content of the paper based on your valuable comments. We would like to request another review of the manuscript. Thank you for your assistance.
- Introduction
Reviewer's suggestion or questions
Row 54 have active against imipenem….reformulate
Response:
This point has been revised as suggested.
- Results
Reviewer's suggestion or questions
Row 79 Initially, 7,058 clinical specimens were collected in our hospital
Row 112, figure 1- 152+3685=3837, 15+139=154
Response:
We have checked and corrected the numbers.
Reviewer 3 Report
Title: Screening for Metallo-beta-lactamases in Enterobacterales: Utility and Limitations of Non-carbapenem Agents
This manuscript is not ready to publish in the journal as many weak points were presented in it. However, I do believe that if they can improve the manuscripts following all comments. It might have a chance to publish in the journal.
Comments
1. Topic: The title is not clear and should be modified.
2. Line 14: Enterobacterales should be replaced by bacteria
3. Line 14: Infection caused by carbapenem-resistant Enterobacterales (CRE) frequently occur in…..
4. Line 16-17: Please identify the objective of this study. It may be better if the author uses passive voice instead of, we aimed to screen ……
Line 20-21: “Additionally, isolates meeting the criteria for CRE were directly subjected to PCR testing”. Please remove or re-write
5. The result in the abstract should be included the quantitative data such as percentage of the resistance etc.
6. Line 36-37: “MBLs are zinc-type β-lactamases categorized under Class B according to the Ambler classification, the most widely used classification of β-lactamases” This sentence should be the second sentence.
7. Line 35-36: “Enterobacterales that produce carbapenemase are referred to as carbapenemase-producing Enterobacterales (CPE)”. This sentence should be the third sentence.
8. Line 73-75: It may be better if the author uses passive voice instead of, we discuss ……
9. The main results of this study are antibiotic susceptibility test based on the MIC values. Then, the authors presented raw data of the antibiotic susceptibility as Table 1, Fig. 1, Fig. 2, Table 2. It would be better if the author added some experiments to achieve results.
10. I recommend detection of antibiotic resistance genes that are involved in the resistance to the antibiotics.
11. The references of the articles that published in 2020-2023 are suggested to be cited.
12. Please delete some old references.
-
Author Response
Point-by-point response to the Reviewers’ comments
Thank you for taking the time to review and provide guidance on my paper. We have revised the content of the paper based on your valuable comments. We would like to request another review of the manuscript. Thank you for your assistance.
Reviewer 3
Response to Reviewer's Comments:
Thank you for your valuable feedback and suggestions. We appreciate your time and effort in reviewing our manuscript. Please find our responses to each of your comments below:
Reviewer's suggestion or questions
- Topic: The title is not clear and should be modified.
Response:
We have included the research objectives in the title to make it more specific and clearer. This revision allows readers to immediately understand the purpose of the study.
Reviewer's suggestion or questions
- Line 14: Enterobacterales should be replaced by bacteria
Response:
We agree that the use of "Enterobacterales" may not be appropriate in this context. As multiple papers use the term "organism" rather than "bacteria," we have changed this term to "organism" in the instance.
Reviewer's suggestion or questions
- Line 14: Infection caused by carbapenem-resistant Enterobacterales (CRE) frequently occur in…..
Response:
We have revised the text based on your feedback. However, as there is no specific mention of the changes to be made, we would like to request that you review the text again to confirm if the revisions meet your expectations.
Reviewer's suggestion or questions
- Line 16-17: Please identify the objective of this study. It may be better if the author uses passive voice instead of, we aimed to screen ……
Response:
We have clearly stated the research objectives and revised the passive voice, as per your request. Please review the text again to ensure that the objectives are accurately and explicitly described in the desired format.
Reviewer's suggestion or questions
- Line 20-21: “Additionally, isolates meeting the criteria for CRE were directly subjected to PCR testing”. Please remove or re-write
Response:
The context has been revised as suggested.
Reviewer's suggestion or questions
- The result in the abstract should be included the quantitative data such as percentage of the resistance etc.
Response:
We have included specific numerical values such as the number of tests, bacterial counts, and percentages of bacteria in the abstract as suggested.
Reviewer's suggestion or questions
- Line 36-37: “MBLs are zinc-type β-lactamases categorized under Class B according to the Ambler classification, the most widely used classification of β-lactamases” This sentence should be the second sentence.
Response:
The text has been revised as suggested.
Reviewer's suggestion or questions
- Line 35-36: “Enterobacterales that produce carbapenemase are referred to as carbapenemase-producing Enterobacterales (CPE)”. This sentence should be the third sentence.
Response:
The text has been revised as suggested.
Reviewer's suggestion or questions
- Line 73-75: It may be better if the author uses passive voice instead of, we discuss ……
Response:
The text has been revised as suggested.
Reviewer's suggestion or questions
- The main results of this study are antibiotic susceptibility test based on the MIC values. Then, the authors presented raw data of the antibiotic susceptibility as Table 1, Fig. 1, Fig. 2, Table 2. It would be better if the author added some experiments to achieve results.
Response:
Thank you for your comments regarding additional testing. Regarding the detailed MIC, would it be acceptable to focus on the MIC against carbapenems? As you pointed out, this study is based on antimicrobial susceptibility testing. However, an important aspect of this study is screening of the prevalent IMP-type MBL in Japan using automated systems and we consider it important to be able to screen for MBL even in facilities that cannot measure low MICs. Therefore, we did not measure detailed values as it was beyond the scope of this study.
Reviewer's suggestion or questions
- I recommend detection of antibiotic resistance genes that are involved in the resistance to the antibiotics.
Response:
As you pointed out, searching for resistance genes is currently the mainstream approach. However, the purpose of this study was to easily and quickly detect or rule out the presence of MBL-producing bacteria in clinical settings. Thus, we did not search for resistance genes using whole-genome analysis, as this is beyond the scope of the study.
Reviewer's suggestion or questions
- The references of the articles that published in 2020-2023 are suggested to be cited.
- Please delete some old references.
Response:
For the sections after Discussion, we have replaced several references with newer references and removed some outdated references. However, for the Introduction, we have retained the older references, as they provide historical context on resistant bacteria and the papers that formed the basis of this study.
For your convenience, the comments in this file have been translated to Japanese using our state-of-the-art AI-based translation tool (accuracy may not be 100%). You may also refer to the comments in English in this file, as usual.
Reviewer 4 Report
This manuscript aimed to screen and characterize MBL-producing isolates from a university hospital. However, we wish to call the authors' attention to a few observations.
Areas of concern:
Introduction
Lines 69-71: It would have been interesting for the authors to briefly highlight the limitations of the existing MBL screening methods included in the testing protocol for laboratory procedures to justify better the use of ceftazidime and cefoperazone/sulbactam for MBL screening tests in the present study.
Results
Figure 1: read positive 154 isolates instead of 152 in the third box (left).
Discussion
Lines 151-156: This should be taken to the methodology section.
Author Response
Point-by-point response to the Reviewers’ comments
Thank you for taking the time to review and provide guidance on my paper. We have revised the content of the paper based on your valuable comments. We would like to request another review of the manuscript. Thank you for your assistance.
Reviewer 4
Response to Reviewer's Comments:
Thank you for your valuable feedback and suggestions. We appreciate your time and effort in reviewing our manuscript. Please find our responses to each of your comments below:
- Introduction
Reviewer's suggestion or questions
Lines 69-71: It would have been interesting for the authors to briefly highlight the limitations of the existing MBL screening methods included in the testing protocol for laboratory procedures to justify better the use of ceftazidime and cefoperazone/sulbactam for MBL screening tests in the present study.
Response:
As you pointed out, I have described the background for the introduction of screening using ceftazidime and cefoperazone/sulbactam. As mentioned in the text, previously, the selection of targets for carbapenemase confirmation tests was based on manufacturer warnings and the experience of laboratory technicians.
- Results
Reviewer's suggestion or questions
Figure 1: read positive 154 isolates instead of 152 in the third box (left).
Response:
The text has been corrected as suggested.
- Discussion
Reviewer's suggestion or questions
Lines 151-156: This should be taken to the methodology section.
Response:
The text has been moved as suggested.
Round 2
Reviewer 3 Report
I have read it carefully and have decided that the authors have addressed reviewer concerns satisfactorily.
The manuscript and the grammar improved.